# Coercivity Increase of the Recycled HDDR Nd-Fe-B Powders Doped with DyF_3_ and Processed via Spark Plasma Sintering & the Effect of Thermal Treatments

**DOI:** 10.3390/ma12091498

**Published:** 2019-05-08

**Authors:** Awais Ikram, M. Farhan Mehmood, Zoran Samardžija, Richard Stuart Sheridan, Muhammad Awais, Allan Walton, Saso Sturm, Spomenka Kobe, Kristina Žužek Rožman

**Affiliations:** 1Department for Nanostructured Materials, Jožef Stefan Institute, Jamova 39, SI-1000 Ljubljana, Slovenia; farhan@ijs.si (M.F.M.); zoran.samardzija@ijs.si (Z.S.); saso.sturm@ijs.si (S.S.); spomenka.kobe@ijs.si (S.K.); tina.zuzek@ijs.si (K.Ž.R.); 2Jožef Stefan International Postgraduate School, Jamova 39, SI-1000 Ljubljana, Slovenia; 3School of Metallurgy and Materials, University of Birmingham, Edgbaston, Birmingham B15 2TT, UK; R.S.Sheridan.1@bham.ac.uk (R.S.S.); m.awais@bham.ac.uk (M.A.); A.Walton@bham.ac.uk (A.W.)

**Keywords:** rare earth permanent magnets, HDDR Nd_2_Fe_14_B, recycling, spark plasma sintering, coercivity, doping DyF_3_

## Abstract

The magnetic properties of the recycled hydrogenation disproportionation desorption recombination (HDDR) Nd-Fe-B powder, doped with a low weight fraction of DyF_3_ nanoparticles, were investigated. Spark plasma sintering (SPS) was used to consolidate the recycled Nd-Fe-B powder blends containing 1, 2, and 5 wt.% of DyF_3_ grounded powder. Different post-SPS sintering thermal treatment conditions (600, 750, and 900 °C), for a varying amount of time, were studied in view of optimizing the magnetic properties and developing characteristic core-shell microstructure in the HDDR powder. As received, recycled HDDR powder has coercivity (H_Ci_) of 830 kA/m, and as optimally as SPS magnets reach 1160 kA/m, after the thermal treatment. With only 1–2 wt.% blended DyF_3_, the H_Ci_ peaked to 1407 kA/m with the thermal treatment at 750 °C for 1 h. The obtained H_Ci_ values of the blend magnet is ~69.5% higher than the starting recycled HDDR powder and 17.5% higher than the SPS processed magnet annealed at 750 °C for 1 h. Prolonging the thermal treatment time to 6 h and temperature conditions above 900 °C was detrimental to the magnetic properties. About ~2 wt.% DyF_3_ dopant was suitable to develop a uniform core-shell microstructure in the HDDR Nd-Fe-B powder. The Nd-rich phase in the HDDR powder has a slightly different and fluorine rich composition i.e., Nd-O-F_2_ than in the one reported in sintered magnets (Nd-O-F). The composition of reaction zone-phases after the thermal treatment and Dy diffusion was DyF_4_, which is more abundant in 5 wt.% doped samples. Further doping above 2 wt.% DyF_3_ is ineffective in augmenting the coercivity of the recycled HDDR powder, due to the decomposition of the shell structure and formation of non-ferromagnetic rare earth-based complex intermetallic compounds. The DyF_3_ doping is a very effective single step route in a controlled coercivity improvement of the recycled HDDR Nd-Fe-B powder from the end of life magnetic products.

## 1. Introduction

The Nd-Fe-B type permanent magnets are vital from a technological point of view, spanning in applications like: Electric motors, medical imaging, telecommunication, and consumer electronics because of their high energy product [1,2,3]. Since the supply chain of rare earth elements (REE) was plagued with the crisis in 2011 [4], they are effectively considered as the most critical raw materials especially for incessantly growing automotive applications [1,5]. Therefore, to meet this demand, the recycling of the rare earth (RE) based permanent magnet is a feasible option [6,7,8,9,10,11,12,13,14,15,16,17,18], which has proven successful in developing sintered magnets from the end-of-life (EOL) scrap. The hydrogenation disproportionation desorption recombination (HDDR) Nd-Fe-B powder, from the recycled scrap, has been effectively consolidated to fully dense magnets via pulsed electric current sintering technique, retaining the coercivity (H_Ci_) of the EOL magnet [19]. The HDDR powder has less than 30 wt.% of rare earth elements (i.e., RE lean), so modification of the grain boundary structure is necessary for increasing the coercivity [12,20]. The coercivity enhancement helps prevent higher temperature demagnetization in high torque permanent magnet motors [5].

Several researchers have utilized rare earth fluorides (RE-F_3_) as dopant or surface diffusion agents, to increase the coercivity in sintered magnets; with the coercivities reaching up to 2790 kA/m (35 kOe) [21,22,23,24,25,26,27,28,29,30,31,32] and in rapidly consolidated melt-spun ribbons, where coercivities reached up to 1990 kA/m [33,34]. On the contrary, no such research has been made on the HDDR Nd-Fe-B powders, which is a relatively cheap hydrogen reprocessing alternative for the EOL magnetic products [6]. In this study, we compared the magnetic properties and microstructural characteristics after doping the recycled HDDR powder with DyF_3_. These blends are rapidly consolidated with Spark Plasma Sintering (SPS), which has been proven to preserve the microstructure, as well as the magnetic properties of HDDR Nd-Fe-B powder. The magnetic properties of DyF_3_ blended recycled HDDR powder are compared with un-doped powder, consolidated with the SPS. The discussion is focused on the mechanism of diffusion at different thermal treatment conditions and the formation of core-shell structure in the HDDR Nd-Fe-B system.

## 2. Experimental

The recycled HDDR powder, as used in the previous study [19], has a nominal composition: Nd_13.4_Dy_0.67_Fe_78.6_B_6.19_Nb_0.43_Al_0.72_. The oxygen content of 4760 ppm was measured with Eltra ON 900 oxygen and nitrogen analyzer (Haan, Germany). The DyF_3_ powder was grounded by a mortar and mixed in weight fractions of 1, 2 and 5% with the recycled HDDR powder.

A total 3 g of the powder blend was added to 10 mm graphite dies, with spacers on the top and bottom. The dies were sealed in a vacuum bag within the glove box to avoid oxidation during the sample handling. A uniaxial pressure of 5 kN was applied to squeeze the powder blend. No prior magnetic alignment was made on the blended powder. Under protective Ar gas, the graphite dies, containing blends, were placed inside the Syntex 3000 (Fuji DR. SINTER, Saitama, Japan) SPS furnace with a uniaxial pressure controller. The SPS operation was performed under 2 × 10^−2^ mbar vacuum. The sintering temperature was kept at 750 °C, which was optimized from the previous study and 100 MPa uniaxial pressure was constantly applied. The heating rate of 100 °C/min was maintained until 700 °C and reduced to 50 °C/min for reaching the maximum temperature; 1 min of holding time was given at 750 °C to reach nearly full densification. The sintering temperature was measured with a calibrated infrared pyrometer. After the SPS operation, the samples were grinded with SiC papers to remove the graphite spacers. The demagnetization measurements were taken on a permeameter (Magnet-Physik Dr. Steingroever, Cologne, Germany). The relative density was measured with bulk density-measurement system (Exelia AG DENSITEC, Zurich, Switzerland) based on Archimedes principle by submerging the samples in silicone oil. 

The thermal treatments were performed at 600, 750, and 900 °C within a horizontal tube furnace under a vacuum of > 10^−5^ mbar with a heating rate of 50 °C/minute. The magnetic measurements were retaken on thermally treated samples. The samples were thermally demagnetized at 400 °C for 15 min in the vacuum furnace. For the microstructural analysis, the demagnetized samples were finely grinded by 2400 grit SiC papers and polished with 1/4 µm diamond paste on a velvet cloth. The microstructural studies were carried out with a Field Emission Scanning Electron Microscope (JEOL 7600F, Tokyo, Japan). The phase identification and elemental distribution were carried out at 20 keV with an electron energy dispersive X-ray spectroscopy (EDXS, Oxford Instruments, High Wycombe, UK) analysis fitted with a 20 mm^2^ Oxford X-Max detector (High Wycombe, UK).

## 3. Results and Discussion

### Characterization of Coarse Recycled HDDR Nd-Fe-B Powder.

Figure 1a shows the initial recycled HDDR Nd-Fe-B powder, with an average particle size of 220 µm. The agglomerates of DyF_3_ powder, shown in Figure 1b. These can be grounded to a fine powder of size <500 nm on average as in Figure 1c before blending with Nd-Fe-B powder. The weight fractions of 1, 2, and 5% DyF_3_ dopant were grounded and then mixed with the recycled HDDR powder. Figure 1d shows fine individual nanoparticles of DyF_3_ uniformly dispersed on the recycled HDDR powder grains of 400 nm average size.

Figure 2 shows the dopant distribution in the back-scattered electron (BSE) microstructure of as SPS samples, with DyF_3_ in different weight fractions. It is quite clear that for 1 wt.% DyF_3_, the microstructure appears quite similar to the representative microstructure of the sintered HDDR powder [19] and the distribution is uniform. With higher content of DyF_3_, the dopant nanoparticles are concentrated along the edges of the HDDR powder particles and also the bright contrast increased towards the center of the particle. In the case of 1 and 2 wt.% DyF_3_, the dopant clustering (<5 µm) was not very prominent and its distribution along the HDDR particles can be considered as uniform. The dopant agglomerates in size range, from 10–100 µm, were observed for 5 wt.% DyF_3_ SPS sample, as shown in Figure 2c. The presence of non-ferromagnetic phase was projected to reduce the sintered density, as well as remanence [34].

Figure 3 shows the magnetic properties of DyF_3_ doped recycled HDDR powder, consolidated with SPS at 750 °C for 1 min and then vacuum heat-treated at 600 °C for 6 h to compare the annealing parameters from the literature [33,34]. The blended magnets with 1, 2 and 5 wt.% DyF_3_ after annealing at 600 °C, resulted in coercivity values of 1274, 1252, and 201 kA/m, respectively. Since no prior magnetic alignment of the blend powder was performed before the sintering, the presented results of remanence conclude that the samples were isotropic. The B_r_ was 0.78 T for 1 wt.% dopant and reduced to 0.75 T for 2 wt.%, and then dropped down substantially to 0.53 T for 5 wt.% DyF_3_. The un-doped recycled HDDR powder has H_Ci_ = 830 kA/m and B_r_ = 0.92 T. Using similar SPS reprocessing conditions, the sintered magnets from the un-doped recycled HDDR powder resulted in H_Ci_ = 1060 kA/m and B_r_ = 0.76 T, which improved to H_Ci_ = 1160 kA/m and B_r_ = 0.77 T after vacuum heat treatment at 750 °C [19]. 

The melting temperature (T_m_) for DyF_3_ is 1350 °C and previous reports on doped sintered Nd-Fe-B magnets suggested a high thermal treatment temperature range (800–950 °C) [26,28,34]. The eutectic liquid phase transformation in the HDDR system begins at 665 °C and transformation is complete at 743 °C [19]. Thereby, in order to determine the magnetic property variation, temperatures of 750 °C (above the eutectic transformation temperature) and 900 °C (above the HDDR Nd-Fe-B grain growth temperature [35,36]) were selected. As the annealing temperature was increased to 750 and 900 °C, the holding time was varied from 1 to 6 h to identify the changes in magnetic properties. In the thermal treatment at 900 °C, the coercivity, as shown in Figure 4a, reached a maximum value of 1279 kA/m for 2 wt.% doped magnets thermally treated for a relatively shorter timeframe (1 h). By increasing the dopant weight fraction as well as holding time, the coercivity declined substantially to 452 kA/m (5 wt.% DyF_3_ for 6 h). The B_r_ dropped from 0.77 T to 0.72 T for 1 wt.% dopant after annealing for 1 and 6 h respectively. For higher weight fraction of dopant, the reduction in B_r_ was more significant after thermal treatments at 900 °C as compared to 750 °C sample as shown in Figure 4b. 

Figure 4c shows the coercivity increased to 1407, 1399, and 1212 kA/m for 1, 2, and 5 wt.% DyF_3_ respectively after the heat treatment at 750 °C for 1 h. The coercivity began to decline by increasing the holding time from 1 to 3, and 6 h. The coercivity for 1 and 2 wt.% DyF_3_ remained over ~1200 kA/m even after 6 h of thermal treatment, whereby it declined sharply to 977 kA/m for 5 wt.% sample held for 6 h. The H_Ci_ of 2 wt.% DyF_3_ was slightly higher when heat treatment was increased to 3 h or more as compared to 1wt.% doped samples. The B_r_ increased from 0.75T to 0.85 T for 1 wt.% DyF_3_ blend sample after 1-h thermal treatment. The B_r_ was in the range of 0.77–0.78 T after the heat treatment up to 6 h for 1 and 2 wt.% DyF_3_, as shown in Figure 4d. The B_r_ values were gradually reduced from 0.77 T to 0.70 T as the holding time was increased from 1 to 6 h in the 5 wt.% doped samples.

Figure 5 shows the SEM microstructure in backscattered mode after the thermal treatment at 750 °C for 6 h of 1–5 wt.% SPS-ed HDDR powder. As compared with Figure 2, with the thermal treatment, there has been an evident formation of core-shell structure. At low dopant weight fraction (1%), the microstructure consists of two zones as shown in Figure 5a,b: Zone A consists of a core-shell structure and zone B is characteristic of HDDR powder i.e., a thin grain boundary film between the matrix grains. The core shell microstructure is more homogenous with an increase in dopant content up to 2 wt.% as shown in Figure 5c,d. As the DyF_3_ weight concentration is increased to 5%, the microstructural homogeneity was reduced. Figure 5e shows a core-shell zone with the unreacted and non-diffused dopant (bright phase). This microstructure was not homogenous for the 5 wt.% doped samples as Figure 5f shows an abnormal growth zone with excessive formation of DyNdFe_14_B shells over the Nd_2_Fe_14_B matrix grains. This abnormal grain coarsening in 5 wt.% doped samples also reflected in significantly poor magnetic properties (H_Ci_ = 977 kA/m and B_r_ = 0.7 T).

After thermal treatment, the region of dopant nanoparticles becomes the reaction zones (RZ), from where the Dy diffuses with the liquid phase towards the center of the particle and precipitate on the surface of the matrix grains, as shown in Figure 6. The morphology and phase formation of bright phases change with an increase in dopant concentration. The reaction zones are smaller in 1 wt.% samples, as in Figure 6a, where the Dy diffusion is limited to matrix phase in the vicinity of the reaction zone. The 2 wt.% DyF_3_ sample has the most optimal microstructure and magnetic properties after the thermal treatment at 750 °C, the reaction zone widens and the formation of complex interphases was identified with EDXS in Figure 6b. The phase No. 1 corresponds to the Nd_2_Fe_14_B matrix phase and after the Dy diffusion, the shell structure of DyNdFe_14_B reinforces the matrix phase. The shell structure has ~6 at.% Dy as compared to the cores with ≤ 1.2 at.% of Dy. The nominal phase composition of the core-shell structure measured with EDS is presented in Table 1. Before thermal treatment, the DyF_3_ nanoparticles do not react or decompose and the Nd-rich phase composition corresponds to NdO_X_ (No. 2). The Nd_2_O_3_ oxide phase (No. 3), NbFe_2_ Laves (No. 4), and the tetragonal NdFe_4_B_4_ boride (No. 9) phases, were initially present in the recycled HDDR powder. From Figure 6b, the primary Nd-rich phase identified (No. 12) in the HDDR Nd-Fe-B system has fluorine-rich composition Nd-O-F_2_. Similar to the conventionally sintered magnets and melt-spun ribbons, the oxyfluoride phases Nd-O-F (No. 5) and Dy-O-F (No. 6) were also detected along the RZ. The complex interphase compounds were easy to analyze in the relatively larger RZ of the 5 wt.% doped sample, as shown in Figure 6 c–f. The Nd-rich oxyfluoride regions Nd-O-F and Nd-O-F_2_ had similar greyish contrast and could only be identified with the EDXS. After the thermal treatment the constituent dopant particle has a composition of DyF_4_ (No. 10) as shown in Figure 6c. As the core-shell structure forms up and Dy begins to diffuse further from the RZ towards the HDDR particles, this phase is reduced to NdF_4_ (No. 7). The interphase complexes are shown in Figure 6e,f surrounding the RE-F_4_ phase (dopant particle) are listed in Table 1 as Nd-Fe-O-F (No. 8), DyFe_2_ (No. 11), Nd-Fe-F (No. 13 and 15) and Dy-Nd-O-F_2_ (No. 14). During the prolonged thermal treatment (750 °C–6 h), the core-shell structure in 5 wt.% sample is obvious in regions closer to the RZ (Figure 6e), whereby the matrix structure collapses, due to the exaggerated growth of shells, as shown in Figure 5f. The presence of porosity in the RZ of Figure 6f indicates the diffusion of Dy until the core-shells grow throughout the microstructure, which gets circumvented as the diffusion stops and the complex RE-Fe-F intermetallic phases begin to form (No. 11, 13 and 15). 

## 4. Discussion

The concept of recycling the permanent magnets includes the techniques like: hydrometallurgical, pyro-metallurgical, direct reusage and indirect recycling approaches [37,38,39,40,41,42,43,44,45,46]. The advantage of direct recycling and re-usage methods, based on hydrogen-based technologies and sintering, which we have proposed in our previous study [19] and the present work, is that they have a smaller environmental footprint, as compared to conventional hydro- and pyro-metallurgical methods, which are energy intensive and require plentiful of highly corrosive chemical mediums. Therefore, the hydrogen-based methods offer more economical and energy efficient route to obtain pulverized and demagnetized powder from the end-of-life (EOL) magnets. The added benefit of hydrogenation disproportionation desorption recombination route is that the powder with anisotropic nano-crystalline grains can be used as plastic bonded as well as sintered magnets.

The rapidly sintered magnets from the recycled HDDR powder were prepared by blending low weight fraction of fine DyF_3_ powder, by rapidly consolidating it with spark plasma sintering and thermally treating at different conditions. The recycled HDDR powder has a nominal grain size of 240–420 nm, which is slightly above the critical single domain size. Developing this characteristic microstructure from the EOL magnet is vital during the HDDR reprocessing for achieving the desired level of magnetic properties is important. The suitable reuse of reprocessed powder can be difficult if the scrap magnet has already been excessively oxidized or corroded in the harsh environments. The recovery of magnetic properties is strongly dependent on the initial chemical composition of the EOL magnets and recycling different magnets in a single batch, which were used in different service conditions, is challenging. Nonetheless the rejected industrial waste or the EOL material from similar appliances can be effectively recycled by the hydrogen processing routes. Practically, it is difficult to increase the coercivity of the as prepared recycled HDDR powder, if the EOL magnets do not contain an excessive amount of heavy rare earth elements (HREE: Dy, Tb). It is also not technologically feasible to add the HREEs in multiple stages of the HDDR process. Contemporary solution for increasing the H_Ci_ of the recycled HDDR powder is by blending small amounts (1–5 wt.%) of DyF_3_ powder and SPS to bulk magnets, followed by the experimentally-determined optimal heat treatment at 750 °C, increasing the H_Ci_ > 1400 kA/m, which is 69.5% higher than the starting recycled HDDR powder as shown in Figure 3 and Figure 4. The H_Ci_ increment is more prominent at 1–2 wt.% dopant addition and subsequent thermal treatment above eutectic liquidus transition temperature i.e., 750 °C for shorter periods, due to the formation of uniform core-shell microstructure. It is relatively easy for Dy to diffuse at temperatures above the eutectic transition at 665 °C. The activation energy for decomposition of heavy rare earth based fluorides like DyF_3_ is lower than NdF_3_, which stimulates the high diffusivity of Dy via Nd-rich grain boundary channels, from the edges of the HDDR particles to the center during the thermal treatments [22].

According to U. M. R. Seelam et al. [36], in vapor sorption treated conventionally sintered magnets, the Dy vapors become part of the liquid phase at 900 °C (above the ternary transformation temperature) and the Dy-rich (DyNd)_2_Fe_14_B shells precipitate out of the liquid phase upon cooling and condense on the surface of 2:14:1 grains. At 900 °C, the surface of Nd_2_Fe_14_B grains experience partial melting and more Nd atoms become part of this Nd/Dy rich liquid phase. Since the sintered magnets have single, crystal Nd_2_Fe_14_B grains, in the size range of 5–10 µm, and the continuous grain boundary phase surrounding all the matrix grains, a high Dy concentration at the GBs during sorption treatment effectively develops the (DyNd)_2_Fe_14_B core-shell structure. The core-shell formation mechanism, as proposed by U. Seelam [36], is based on the solidification of Dy-enriched liquid Nd_2_Fe_14_B that partially melted upon annealing and the Nd-rich phase, which connects all the grains in the sintered magnet. The (DyNd)_2_Fe_14_B shell begins to develop coherently below 900 °C on the Nd_2_Fe_14_B facets as the Nd-rich phase is still in a liquid state above 665 °C. Below this ternary transition temperature ~400 nm thick DyNdFe_14_B shells already have deposited on the surface of Nd_2_Fe_14_B grains and the solidifying liquid phase now contains a lesser amount of Dy, but a higher concentration of Nd. Since the formation energies of Dy_2_Fe_14_B structure is more negative (favorable) than Nd_2_Fe_14_B, a preferential solidification proposes DyNdFe_14_B phase to solidify on the facets of partially melted Nd_2_Fe_14_B grains. Dy remains part of the liquid phase until solidification below 700 °C. With negative enthalpy of formation, DyNdFe_14_B shell will form the first along the partially molten surface of the Nd_2_Fe_14_B grains. 

In the case of the recycled HDDR blended with the DyF_3_ nanoparticles, the explanation of complex diffusion process may not adhere entirely to this theory. The recycled HDDR powder particle are sized on average 220 µm, whereas each particle is composed of networked ~400 nm sized nanocrystalline grains. The Nd-content of the recycled HDDR powder is lean, as from the previous studies, due to higher oxygen content and the multiple grains are sometimes even in direct contact with each other, due to the low amount and lack of the grain boundary phase in some places [19]. Since the grain boundary thickness is not more than 3 nm in the HDDR system, the capillary forces are responsible for the liquid phase transport and uniform dispersal along the 2:14:1 grains [10,47,48].

The mechanism for subtle H_Ci_ increment with the low weight fraction of dopant in the recycled HDDR based magnets and subsequent degradation of magnetic properties for ≥ 5 wt.% DyF_3_, containing samples, is characterized with the aid of EDS composition analysis for various thermal treatment conditions. After SPS reprocessing, diffusion of Dy was limited, as shown in Figure 2 and up to 2 wt.%, the dopant nanoparticles remain finely distributed along the HDDR particles whereby for 5 wt.% dopant, the larger DyF_3_ nanoparticles were segregated along the HDDR particles. Since the optimal SPS reprocessing temperature was 750 °C, short range diffusion of Dy can be expected [33,34] during the liquid phase sintering, allowing the transformation of Nd-rich phase to Nd oxyfluorides, but the typical (DyNd)_2_Fe_14_B core-shell structure was not observed prior to the thermal treatment. Pressure during SPS consolidation aids in the uniform dispersal of dopant nanoparticles [19] and increases their surface distribution with the HDDR powder particles. The short SPS holding time is 1 min, however this restricts any diffusion process and rapid cooling retains the short range order of the dopant nanoparticles, settled along the HDDR powder particles. During the SPS, the only expected transformation is the partial formation of Nd-oxyfluorides along the particle boundaries. However, when Nd-rich liquid phase comes in contact with DyF_3_ nanoparticles above 665 °C, it should decompose and form Nd-oxyfluorides and Dy will become part of the liquid phase. The enthalpy of formation NdF_X_ (X = 3 & 4) is higher than DyF_3_ [33,34]. On the other hand, as the heating and cooling are rapid, the diffusive transport of Dy/Nd-rich liquid phase is restricted during the SPS [19].

During the post-SPS thermal treatment at 600 °C, a partial decomposition of DyF_3_ restricts the formation of the core-shell structure, even at prolonged time periods and H_Ci_ increment is ~6% approx. This indicates the thermal decomposition is favored for substitution between Nd and Dy even at 600 °C [33]. Therefore, the thermal treatment at 600 °C results in only ~70 kA/m improvement of coercivity. With the availability of liquefied Nd-rich phase above the ternary eutectic temperature 665 °C, the Dy gets transported from the edges of the particles towards the center and (DyNd)_2_Fe_14_B shells form upon cooling below 700 °C. Therefore, during the thermal treatment at 750 °C (which is above the ternary eutectic temperature), the DyF_3_ particle decomposes at this temperature and the surface of Nd-Fe-B grain experience melting. Now the HDDR particles ~220 microns in size have the dopant particles sitting at their edges only. The Nd-rich phase is in a liquid state above the ternary eutectic point and connects the GBs to the edges of the particles. In order to form uniform core-shell facets, which are attached to the GBs, liquid phase diffusion is necessary. The decomposed Dy becomes part of the liquid phase and is transported via capillary forces towards the GB interface of nanocrystalline grains. During this thermal treatment, the overall RE-content of GBs is higher than the starting HDDR powder and the liquid phase contains both, Dy atoms from the dopant, as well as the Nd atoms from the partially decomposed grain surfaces. The formation enthalpy of (DyNd)_2_Fe_14_B grains is higher than Nd_2_Fe_14_B grains, so when the cooling begins, the shells are more thermodynamically favorable to precipitate out, as compared to the original Nd_2_Fe_14_B composition on the activated surfaces [36]. At the onset of cooling, and when the 2:14:1 phase begins to condense, the Dy atoms from the liquid phase partially substitute the Nd atoms on the surface of matrix grains (favorable thermodynamic kinetics) and precipitate out as the (DyNd)_2_Fe_14_B shell structure [33] of thickness ≤100 nm, forming thoroughly in 2 wt.% doped samples. The diffusion of Dy atoms is not localized at the particle edges as in case of 600 °C heat treated samples and rapidly diffuses via GB channels to all the intergranular regions of the individual HDDR particle. Above the eutectic transition temperature (665 °C), the three processes in chronological sequence explain the core-shell formation mechanism are: liquid phase diffusion increase in GB RE-content), upon cooling the Dy-Nd substitution and the (DyNd)_2_Fe_14_B shells precipitation from the solution are definitive in the HDDR Nd-Fe-B system. The B_r_ trend indicates the optimal presence of hard magnetic phase Nd_2_Fe_14_B and DyNd_2_Fe_14_B for up to 2 wt.% of the dopant. The B_r_ decline can be attributed to the formation of non-ferromagnetic intermetallic compounds in 5 wt.% samples. 

Nd is found in the intergranular regions to combine with fluorine and oxygen to form oxyfluorides of composition Nd-O-F_2_. For shorter thermal treatment time at 750 °C in 1 wt.% DyF_3_ sample, the coercivity reaches 1400 kA/m, due to the diffusion of Dy which was found to degrade over prolonged treatments. A partial decomposition of the shell structure is observed, thereby developing two distinct zones in the microstructure, as shown in Figure 5 (A: core-shell near the reaction zone and B: normal HDDR microstructure). The excessive Dy is not available in 1 wt.% samples in forming complex interphase compounds near the reaction zone. With the increase in dopant concentration at 5 wt.%, the reaction zone widens and the intermetallic species (DyFe_2_ and RE-Fe-F), as analyzed in EDS Table 1 originate due to concentration gradient of Dy from the grain boundaries towards the activated facets of Nd_2_Fe_14_B matrix grains. 

The lower activation energy for the thermal decomposition of DyF_3_, above the eutectic transformation temperature, promotes the liquid phase Dy diffusion and the partial substitution of Nd atoms from the activated surfaces of the matrix phase; whereby the highly reactive residual fluorine atoms preferentially react with the Nd-rich phase, forming a higher stability cubic Nd-O-F phase [22]. The intergranular phase of the starting HDDR powder is composed of Nd-rich metallic phase and the NdO_2_ phase with Nd_2_O_3_ (cubic + hcp) oxide phase [19]. The diffused, yet unreacted, F atoms promote the formation of cubic Nd-O-F phase from NdO_2_ phase and the oxyfluorides phase has a smaller lattice mismatch of 2.62% with Nd_2_Fe_14_B matrix, as compared to cubic Nd_2_O_3_ with 3.11%, therefore the smaller the lattice mismatch, the more positive the influence on the H_Ci_ [22]. The hcp-Nd_2_O_3_ having a lattice mismatch of 12.1% with the matrix phase develops micro-strains at the interface and reduce the H_Ci_ [49], remains stable and does not react anymore as it was traced in the doped samples. For greater than 1 wt.% DyF_3_, the main intergranular phase was composed of Nd-O-F_2_ due to excessive fluoride ions available to react with the Nd-rich phase as Dy is consumed in shell formation. The stoichiometric 1:1:1 type oxyfluorides RE-O-F (RE = Dy, Nd) replaced the NdO_2_ phase, whereby the oxyfluorides composition changes to 1:1:2 and Dy-Nd-O-F_2_ (interphase) species accommodating more F in the sample with ≥ 2 wt.% dopant. The DyF_3_ readily decomposes during the thermal treatment even for a shorter holding time of 1 h at 750 °C, allowing complete liquid phase diffusion of Dy via grain boundaries throughout the bulk microstructure and developing a core-shell structure. The liquefied GB, due to the diffusion process are enriched with Dy atoms, which will precipitate in shells on the grains’ surface upon cooling and, in turn, the substituted Nd atoms will become part of the intergranular phase. Therefore, in Figure 5b,d the core-shells structures are separated by continuous thin grain boundary layers. So the overall system should not be lean of Nd-content anymore. Therefore, the core-shell formation and supplication of Nd-rich grain boundary phase in turn increases the coercivity by 69.5%. 

At 5 wt.% dopant, the formation of DyF_4_, NdF_4_ and Nd-Fe-F phases at the triple point regions and particle boundaries increases. The neodymium trifluoride (NdF_3_) having higher formation enthalpy of −1713 kJ/mol, compared to DyF_3_ (−1678 kJ/mol), and has been reported to reduce the magnetic properties as the intermetallic compounds have higher chemical and thermal stability than dysprosium trifluoride (DyF_3_), and form barriers against the Dy diffusion with the liquid phase [33], as shown in Figure 6. The report suggests lanthanide (Nd and Dy) tetrafluorides formed during annealing treatment are more thermally stable than DyF_3_ compound [50], which is a plausible reason for their presence in higher dopant concentration as more Dy diffuses from the DyF_3_ particle towards the matrix, DyF_4_ becomes the composition of residual dopant particle. After the thermal decomposition of DyF_3_ dopant, the shell structure is formed by partial substitution of the Nd atoms by the Dy atoms from the RE-rich liquid phase, which precipitates on the activated surface of Nd_2_Fe_14_B grains and the mechanism is illustrated in Figure 7. During the SPS reprocessing minimal DyF_3_ decomposition or Dy diffusion is expected due to rapid consolidation phenomenon as can be seen in Figure 2. When the thermal treatment begins, the dopant decomposes and Dy atoms become part of the liquid phase. Above the ternary transition temperature, the surface of Nd_2_Fe_14_B grains partially experience melting and Nd atoms infiltrate the Dy-enriched grain boundaries. Upon cooling below 700 °C, the (DyNd)_2_Fe_14_B shells precipitate on the activated Nd_2_Fe_14_B grains’ surfaces as more energetically favorable phase transformation. Finally, below the eutectic point, the excessive Nd atoms become part of the Nd-rich grain boundary phase and, therefore, the GBs are continuous and more uniformly distributed, as compared to the starting recycled HDDR powder. The formation of high anisotropy field (DyNd)_2_Fe_14_B shells and the continuous Nd-rich GB layer surrounding these grain act as a spacer phase to effectively reduce the localized exchange effects, and therefore these two reasons can be associated with the H_Ci_ increment by ~69.5% over the recycled HDDR powder.

A larger concentration of fluoride anions reacts with this metallic Nd (reduced) and NdO_2_, leading to the formation of NdF_4_, and Nd-O-F_2_, respectively, as shown in Figure 6. The EDS results in Table 1 indicated that, with the increase in dopant weight fraction, the concentration of unreacted DyF_4_, intergranular NdF_4_, and RE-Fe-F interphase compounds increase exponentially, which contribute to a significant reduction in the magnetic properties, as shown in Figure 4. The prolongation of thermal treatment at 750 or 900 °C to 6 h also deteriorate the magnetic properties because of the thermally induced disintegration of the core—shell structure as in Figure 5f. As Dy forms more 2:14:1 structure, the substituted Nd is forced to react with highly reactive and mobile F ions in the vicinity, and subsequently, the total amount of Nd_2_Fe_14_B phase is reduced in the system. 

The results of the present study classify the addition of DyF_3_ in the recycled HDDR Nd-Fe-B system shows to be very effective for forming the (DyNd)_2_Fe_14_B core-shell structure. But there is an effective limit of DyF_3_ doping to ≤ 2 wt.%, which not only gives the best magnetic properties, but it is also pivotal to REE criticality and conservation [1,2]. The controlled post-SPS thermal treatment in recycled HDDR system is very efficient for shorter durations and lower temperatures in developing higher coercivity and the core-shell structure, as compared to conventionally sintered magnets treated at elevated temperatures [21,22,23] or prolonged periods [33], due to ultrafine microstructure. For shorter periods, the formation of smaller RZ is important as only Dy forms core-shell structure by Nd substitution to Nd-O-F phase. With excess of F anions, the cubic intergranular phase transforms to Nd-O-F_2_ structure. The DyF_3_ concentration of >5 wt.% contributes to the formation of complex intermetallic compounds with REE (Dy, Nd) as well as stable non-decomposing trifluorides and tetrafluorides (NdF_3_ and NdF_4_) at the expense of the Nd_2_Fe_14_B phase as the Dy diffusion is circumvented at large RZs, which plagues the microstructure with excessive non-ferromagnetic species. 

## 5. Conclusions

The DyF_3_ powder was blended with the recycled HDDR powder prior to SPS reprocessing, in order to determine the variation in magnetic properties and to investigate the microstructural evolution. As SPS blend samples had the magnetic properties lower than the starting HDDR powder, which increased significantly with annealing. The thermal treatment conditions were varied from 600 to 900 °C and from 1 to 6 h. The best annealing conditions were determined for the recycled HDDR powder, i.e., 750 °C for 1 h. By annealing at 750 °C for 1 h, even with 1 wt.% DyF_3_, the coercivity (H_Ci_) of sintered magnets can be increased to 17.5%, as compared with the undoped magnets. Up to 2 wt.% DyF_3_, these coercivity value ~1400 kA/m are ~69.5% higher than the starting recycled HDDR powder. Additionally, the reduction in remanence (B_r_) was insignificant up to 2 wt.% DyF_3_ for thermal treatment at 750 °C for 3 h. Prolonging the thermal treatment time to 6 h causes a reduction in the magnetic properties. The thermal treatment at 900 °C for shorter intervals (1 h) resulted in H_Ci_ = 1280 kA/m in 2 wt.% DyF_3_ samples and above 1200 kA/m for all doping conditions. With a further increase in holding time to 6 h at 900 °C, the magnetic properties decline rapidly to: H_Ci_ = 726 and B_r_ = 0.66 T, which is associated with the formation of NdF_4_ and non-ferromagnetic interphase compounds. By optimal thermal treatments, the microstructure of DyF_3_ doped recycled HDDR based magnets exhibit core-shell structure (DyNdFe_14_B shell and Nd_2_Fe_14_B core). Optimal core-shell formation occurs at 1–2 wt.% DyF_3_ and above this concentration, the Dy diffusion in the microstructure from the liquid phase is heterogeneous leading to the formation of complex intermetallic phases at the reaction zones. The secondary phase composition was identified as: Nd-O-F_2_. Moreover, the Nd_2_O_3_ and RE-O-F phases were also observed, due to the transformation of NdO_X_ type Nd-rich phase in the recycled HDDR powder. For the higher weight fraction of dopant, the DyF_4_ is the main composition of the undiffused aggregates, even after prolonged thermal treatments. Therefore, for very small weight fractions, keeping the rare earth’s criticality and recyclability in perspective, the DyF_3_ doping of the recycled HDDR powder is very effective in forming a uniform core shell microstructure in the sintered magnets, which is indicated by a marked improvement in the magnetic properties. 

## Figures and Tables

**Figure 1 materials-12-01498-f001:**
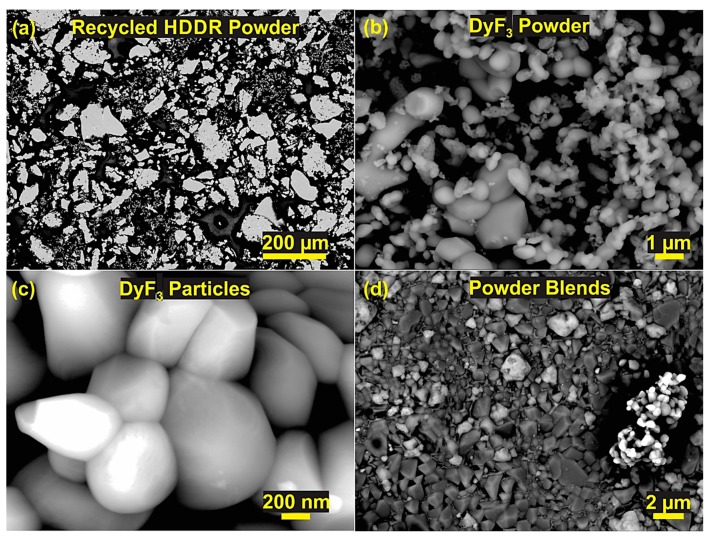
SEM backscattered mode images of (**a**) Hydrogenation disproportionation desorption recombination (HDDR) powder particles, (**b**) DyF_3_ nanoparticles, (**c**) DyF_3_ grains and (**d**) recycled HDDR powder blended with DyF_3_ nanoparticles.

**Figure 2 materials-12-01498-f002:**
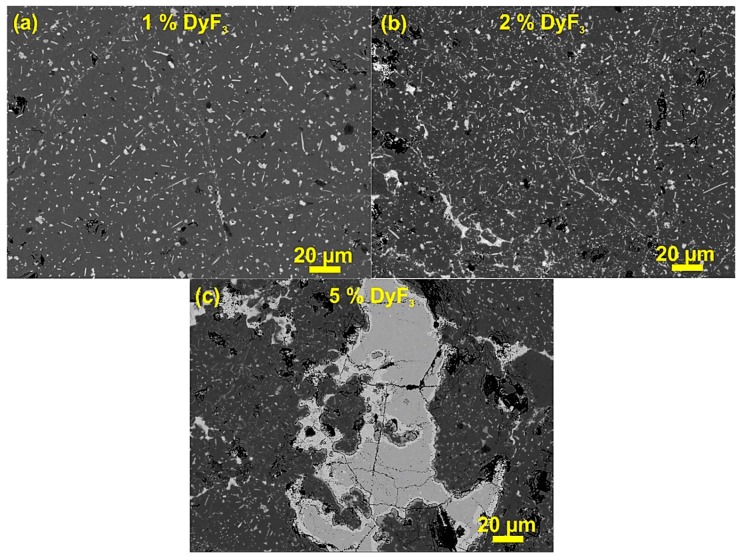
SEM backscattered electron (BSE) images of as-SPS samples with DyF_3_ in (**a**) 1 wt.% (**b**) 2 wt.% and (**c**) 5 wt.%.

**Figure 3 materials-12-01498-f003:**
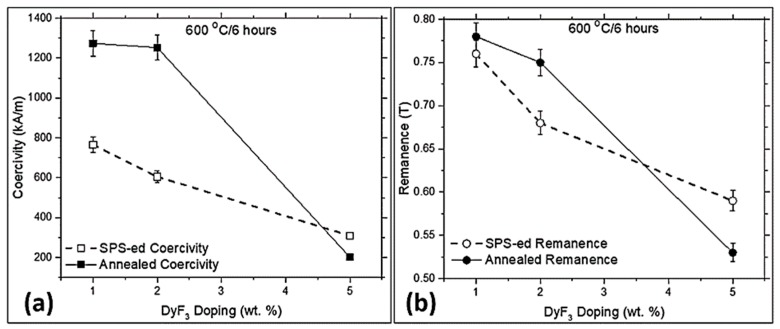
Shows the magnetic properties in DyF_3_ doped magnets before and after annealing at 600 °C for 6 h, (**a**) coercivity and (**b**) remanence.

**Figure 4 materials-12-01498-f004:**
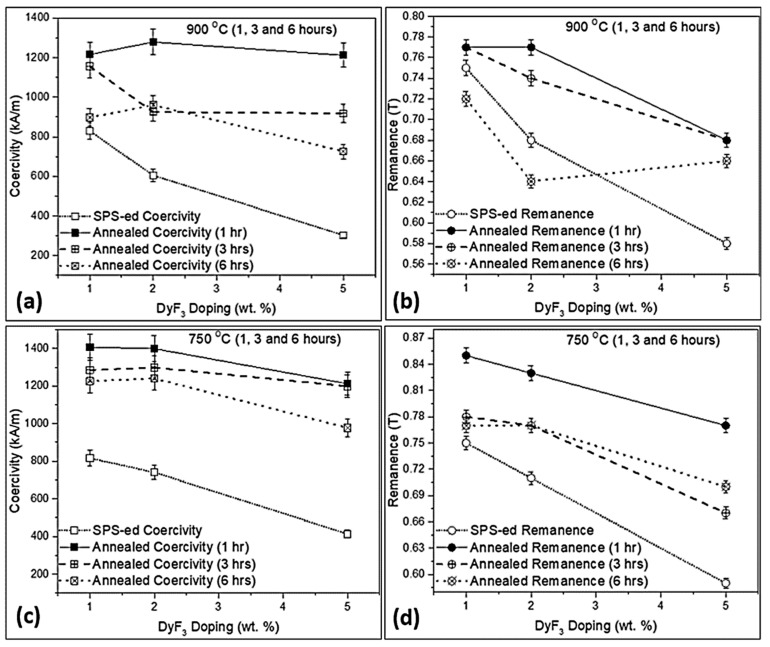
Variation in the magnetic properties of SPS reprocessed blend of DyF_3_ doped recycled HDDR powder with thermal treatment temperatures of 900 °C (**a**) coercivity, (**b**) remanence; and 750 °C (**c**) coercivity, (**d**) remanence.

**Figure 5 materials-12-01498-f005:**
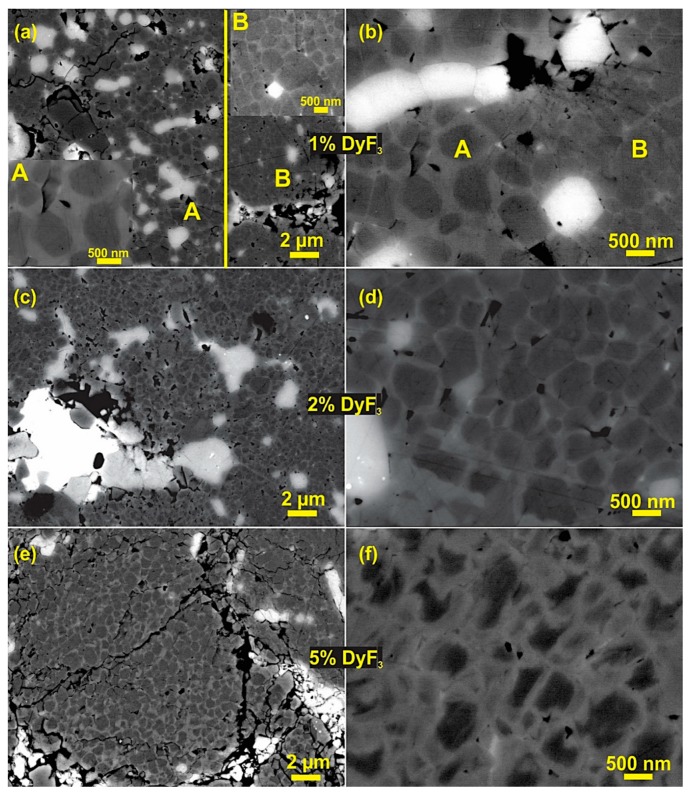
Shows the microstructure of DyF_3_ doped samples after thermal treatment for 6 h at 750 °C, (**a**) 1 wt.% doped samples with two zone microstructure (inset A shows core-shell zone and inset B shows normal HDDR microstructure), (**b**) at higher magnification 1% doped samples; (**c**) 2% DyF_3_ blend samples and (**d**) uniform core-shell structure formation throughout the microstructure; (**e**) 5 wt.% DyF_3_ samples, with excessive growth zone of DyNdFe_14_B shells at the expense of matrix phase clearly shown in (**f**).

**Figure 6 materials-12-01498-f006:**
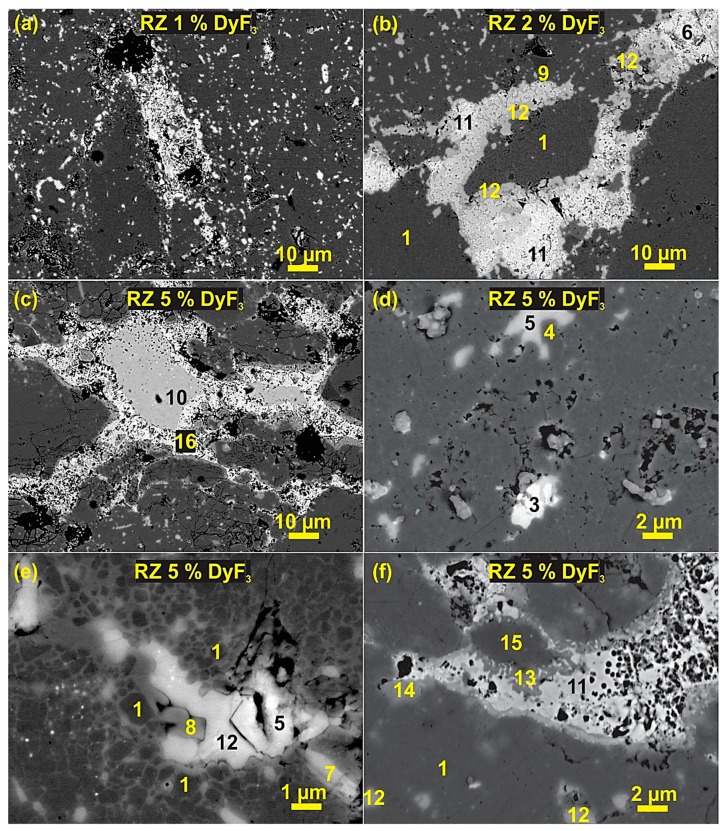
Signifies the reaction zones (RZ) in the microstructure of doped samples after the thermal treatment at 750 °C for 6 h, (**a**) relatively small RZ in 1 wt.% DyF_3_ samples, (**b**) 2 wt.% doped samples have the optimal magnetic properties but RZ contains RE-F_4_ (rare earth fluoride) and Nd-O-F_2_ (oxyfluoride) phases, (**c**–**f**) the relatively wider RZs of 5 wt.% doped samples containing additional interphase compounds along with rare earth fluorides and oxyfluoride based Nd-rich phase.

**Figure 7 materials-12-01498-f007:**
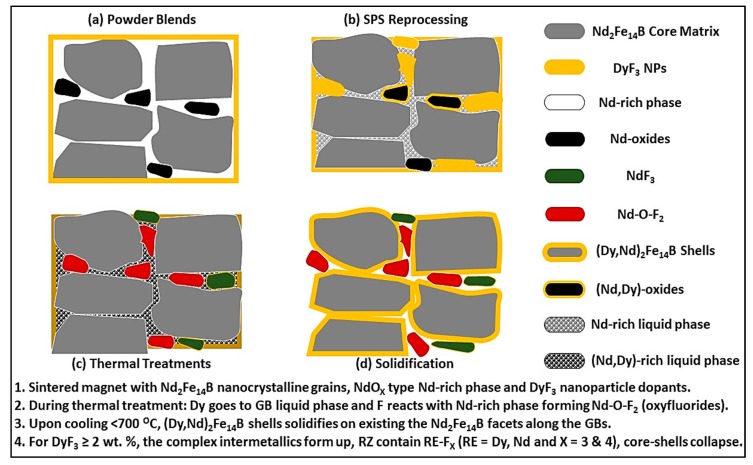
The mechanism of core-shell structure formation in the HDDR Nd-Fe-B system: (**a**) blend of HDDR Nd-Fe-B powder and DyF_3_ particels, (**b**) liquid phase sintering, lack of long range Dy diffusion, (**c**) Dy diffuses with the liquid phase and F reacts with Nd-rich phases; and (**d**) the (Dy,Nd)_2_Fe_14_B core-shell structures form up upon solidification.

**Table 1 materials-12-01498-t001:** EDXS quantification of different phases in the DyF_3_ doped and SPS-ed recycled HDDR Nd-Fe-B.

Phase No.	Phases	Nd (at.%)	Fe (at.%)	F (at.%)	O (at.%)	Nb (at.%)	Al (at.%)	Dy (at.%)
1	Nd_2_Fe_14_B Cores	12	85	-	-	0.9	0.9	1.2
	DyNdFe_14_B Shells	8.1	85.5	-	-	-	-	5.8
2	Nd-rich NdO_x_/NdO_2_	24.6	29.5	-	46.3	-	-	
3	Nd_2_O_3_	34.7	1.8	-	63.5	-	-	
4	NbFe_2_ Laves	0.7	47.6	-	-	51.7	-	
5	Nd-O-F	28.2	1.8	38.5	31.5	-	-	-
6	Dy-O-F	4.5	18.8	20.9	28.9	-	-	26.9
7	Nd-F_4_	18	-	81.1	-	-	-	0.9
8	Nd-Fe-O-F (interphase)	10.8	45.4	28.7	9	-	-	6.1
9	NdFe_4_B_4_	18.8	68.5	9.6	-	-	-	-
10	Dy-F_4_	-	-	80.8	-	-	-	19.2
11	Dy-Fe_2_ (interphase)	2.4	54.6	4.6	10.4	-	-	28
12	Nd-O-F_2_	20.7	2.4	61.3	13.7	-	-	1.9
13	Nd-Fe-F (interphase)	11.9	47.2	33.3	3.6	-	2.3	1.7
14	Dy-Nd-O-F_2_ (interphase)	8.9	6.8	48.8	21.7	-	-	13.8
15	Nd-Fe-F (interphase)	13.3	39.2	39.3	3.9	-	1.3	3

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
