# Peer review of "Coercivity Increase of the Recycled HDDR Nd-Fe-B Powders Doped with DyF3 and Processed via Spark Plasma Sintering & the Effect of Thermal Treatments"

_materials, 2019, doi:10.3390/ma12091498_

Round 1
Reviewer 1 Report
Title: Coercivity increase of the recycled HDDR Nd-Fe-B powders doped with DyF3 and processed via Spark Plasma Sintering & the effect of thermal treatments
The manuscript presents the study on coercivity propertity of the recycled HDDR Nd-Fe-B powders. The studies was involved in DyF3 doping and processed by Spark Plasma Sintering & the effect of thermal treatments. The DyF3 doping demonstrates a very effective single step route in a controlled coercivity improvement of the recycled HDDR Nd-Fe-B powder from the end of life magnetic products. This is well prepared manuscript which is recommend to be published in the journal ‘materials’.
Some minor comments are:
1. What is ‘HDDR’ (Line 11)? Normally all the abbreviations should be written by full name at the first time in the manuscript.
2. There are many long sentences like this ‘Developing this characteristic microstructure from the EOL magnet is vital during the HDDR reprocessing for achieving the desired level of magnetic properties for suitable reuse which can be difficult due to excessive oxidation or environmental corrosion.’ (Line 217-219), it is better for understanding to change it into some short sentences.
Author Response
We are glad to read positive remarks from the reviewer regarding the research quality and the recommendation of manuscript for publication. Moreover we accept all the comments or suggested corrections by the reviewer and incorporate it in to the manuscript as following:
Reviewer #1:
The manuscript presents the study on coercivity propertity of the recycled HDDR Nd-Fe-B powders. The studies was involved in DyF3 doping and processed by Spark Plasma Sintering & the effect of thermal treatments. The DyF3 doping demonstrates a very effective single step route in a controlled coercivity improvement of the recycled HDDR Nd-Fe-B powder from the end of life magnetic products. This is well prepared manuscript which is recommend to be published in the journal ‘materials’.
Some minor comments are:
1. What is ‘HDDR’ (Line 11)? Normally all the abbreviations should be written by full name at the first time in the manuscript.
We thank the reviewer for this comment and regret missing out the important detail as pointed out. Therefore, the abbreviation of HDDR has been added in line 11 – 12 and 43 – 44.
2. There are many long sentences like this ‘Developing this characteristic microstructure from the EOL magnet is vital during the HDDR reprocessing for achieving the desired level of magnetic properties for suitable reuse which can be difficult due to excessive oxidation or environmental corrosion.’ (Line 217-219), it is better for understanding to change it into some short sentences.
The long sentence has been corrected in the text as following from lines 219 – 222:
“Developing this characteristic microstructure from the EOL magnet is vital during the HDDR reprocessing for achieving the desired level of magnetic properties is important. The suitable reuse of reprocessed powder can be difficult if the scrap magnet has already been excessively oxidized or corroded in the harsh environments.”
Reviewer 2 Report
The authors report on the coercivity increase of the recycled HDDR Nd-Fe-B powders doped with DyF3 implementing spark plasma sintering and thermal treatment.
The manuscript clearly written and can be followed with no issues. Data presented is clearly interpreted and the discussion section is sound, supporting experimental findings. Very minor comments are given below, which are not directly related to the technical content:
- Nd-Fe-B are not vital because of BHmax, but because their Jr and Hc capabilities. BHmax is a consequence of both mentioned properties;
- The theoretical (calculated) density presents some ± value. If it is theoretical, no error should exist;
- Although the term "dynamic vacuum" can be understood, the more appropriate one is pressure;
- When the authors claim that the desired level of properties for suitable reuse can be difficult due to excessive oxidation or environmental corrosion, it should also be mentioned that it depends on the initial chemical composition of the alloy, since HDDR processing conditions are strongly linked to the alloy being processed. In other words, recycling different "raw" materials in the same batch might be a challenged to be solved.
At last, I suggest the authors to include one additional perspective in their work (although clearly not technical). The driving force of recycling was the spike in the price of rare earths. Many initiatives have been initiated, and the present work illustrates well one of the possible routes to be followed. However, no information is provided on how feasible it is economically or, in other words, which market would be touched by it. It is relevant for the academic community also to capture the relevance of a technical topic on its potential economic impact because new developments can arise from that. Therefore, my questions to the authors: i-) how economically feasible do you judge this recycling technology? ii-) which market (and why) it is relevant?
Author Response
Reviewer #2:
The manuscript clearly written and can be followed with no issues. Data presented is clearly interpreted and the discussion section is sound, supporting experimental findings. Very minor comments are given below, which are not directly related to the technical content:
- Nd-Fe-B are not vital because of BHmax, but because their Jr and Hc capabilities. BHmax is a consequence of both mentioned properties;
We completely agree with the reviewer on this comment. Since our focus in this study is related to coercivity increment in isotropic magnets, the BHmax has not been outlined except in the introduction. The changes in remanence however have been incorporated based on the dilution of Nd2Fe14B matrix phase when the higher weight fraction of dopant was used. During thermal treatment non-ferromagnetic phases form up abundantly along the dopant particles, which reduced the magnetization values for > 5 wt. % DyF3 and time more than 3 hours at 750 or 900 OC.
- The theoretical (calculated) density presents some ± value. If it is theoretical, no error should exist;
Affirmative and corrected in the text as following from line 64 – 65:
“The theoretical density of the recycled powder was calculated as 7.57 g/cm3 from weight fraction and elements’ density.”
- Although the term "dynamic vacuum" can be understood, the more appropriate one is pressure;
Thank you for pointing it out. In order to simplify the understanding, the term “dynamic vacuum” from line 72 and 82 have been replaced with the word ‘vacuum’ only.
- When the authors claim that the desired level of properties for suitable reuse can be difficult due to excessive oxidation or environmental corrosion, it should also be mentioned that it depends on the initial chemical composition of the alloy, since HDDR processing conditions are strongly linked to the alloy being processed. In other words, recycling different "raw" materials in the same batch might be a challenged to be solved.
This is a well anticipated comment and arguably the biggest unresolved challenge in the magnet recycling scheme. Text has been added in the manuscript to acknowledge this vital point from line 221 – 224:
“The recovery of magnetic properties is strongly dependent on the initial chemical composition of the EOL magnets and recycling different magnets in a single batch which were used in different service conditions is challenging. Nonetheless the rejected industrial waste or the EOL material from similar appliances can be effectively recycled by the hydrogen processing routes.”
At last, I suggest the authors to include one additional perspective in their work (although clearly not technical). The driving force of recycling was the spike in the price of rare earths. Many initiatives have been initiated, and the present work illustrates well one of the possible routes to be followed. However, no information is provided on how feasible it is economically or, in other words, which market would be touched by it. It is relevant for the academic community also to capture the relevance of a technical topic on its potential economic impact because new developments can arise from that. Therefore, my questions to the authors: i-) how economically feasible do you judge this recycling technology? ii-) which market (and why) it is relevant?
The concept of recycling the permanent magnets has been introduced more than a decade ago. Many techniques have been devised to recycle the rare earth magnets including: hydrometallurgical, pyro-metallurgical, direct reusage and indirect recycling approaches [1 – 12]. The advantage of direct recycling and reusage methods based on hydrogen based technologies and sintering which we have proposed in our previous study [10] and the present work have a smaller environmental footprint as compared to conventional hydro & pyrometallurgical methods which are energy intensive and require plentiful of highly corrosive chemical mediums. Therefore, the hydrogen based methods offer more economical and energy efficient route to obtain pulverized and demagnetized powder from the end-of-life (EOL) magnets. The added benefit of hydrogenation disproportionation desorption recombination (HDDR) route is that the powder with anisotropic nanocrystalline grains can be used as plastic bonded as well as sintered magnets. Now when we add up to 1 – 2 wt. % only RE-F3 in the recycled HDDR powder, it is possible to obtain the coercivity values higher than commercially available HDDR powder which is very promising from the recycled material. The better the initial EOL magnets in terms of chemical composition and magnetic properties can be hydrogenated via HD or HDDR and final powder blending will ensure the coercivity is high without a drastic decrease in magnetization and maximum energy products. It is affirming that these reprocessing routes as reported previously [1, 3, 5, 6, 8] have shown good promise in industrial upscaling. Our method provides a contemporary solution to the nanostructured reprocessed magnetic powders, which cannot be dealt with conventional compaction techniques. With the SPS isotropic or anisotropic (hot deformed) magnets from the recycled HDDR powder can be fabricated, with the possibility of industrial upscaling. So in matter of 10 – 15 years, the recycled magnets are projected to become part of the mainstream applications [2, 4], starting with microelectronics and hybrid/full electric vehicles’ motors and traction components. This is one of biggest opportunities for the European producer of rare earth permanent magnets to explore due to up surging demand in the electric vehicles, green technologies and microelectronics [4, 9].
With the addition of RE-F3 dopants, the Nd-rich phase (NdOX) type is transformed in to cubic Nd-O-FX (X = 1, 2) which will not oxidize to hcp-Nd2O3 either under these sintering or annealing conditions, so the intergranular phase is in a sense protected from degradation (oxidation) during reprocessing. The Nd-O-F phase do not have high degree of lattice mismatch (< 3%) as the hcp-Nd2O3 have >12 % with the Nd2Fe14B matrix phase.
The following text from above has been incorporated in the manuscript from line 215 – 224:
“The concept of recycling the permanent magnets includes the techniques like: hydrometallurgical, pyro-metallurgical, direct reusage and indirect recycling approaches [39 - 49]. The advantage of direct recycling and reusage methods based on hydrogen based technologies and sintering which we have proposed in our previous study [19] and the present work have a smaller environmental footprint as compared to conventional hydro & pyrometallurgical methods which are energy intensive and require plentiful of highly corrosive chemical mediums. Therefore, the hydrogen based methods offer more economical and energy efficient route to obtain pulverized and demagnetized powder from the end-of-life (EOL) magnets. The added benefit of hydrogenation disproportionation desorption recombination (HDDR) route is that the powder with anisotropic nanocrystalline grains can be used as plastic bonded as well as sintered magnets.”
Since our expertise lie with material processing and powder metallurgy, accurate economical predications may not be very easy for us to derive quantitatively. But qualitatively in line with the current state of the art reports on rare earth circular economy and sustainability, the researchers are inching towards high performance reprocessed magnets via compositional and microstructural control. On the contrary it is a difficult choice to include too much economical assessment in the following in the manuscript to avoid lengthening and broaden the concise discussion on doping and reprocessing the recycled HDDR Nd-Fe-B powder. But since recently more elaborate economical and sustainability reports are getting published due to centered interest of the global community, the scientific research on the rare earth EOL magnets’ recycling will intensify to support the commercial prospects. More importantly the coercivity mechanism and atomistic diffusion shall be thoroughly investigated to realize the higher coercivities in near single domain sized nanocrystalline magnetic powders and possibly control the microstructural inhomogeneities.
Additional references 39 – 48 have also been added in the manuscript from line 552 – 580:
39. Y. Yang, A. Walton, R. Sheridan, K. Güth, R. Gauss, O. Gutfleisch, M. Buchert, B. M. Steenari, T. V. Gerven, P. T. Jones, K. Binnemans, REE Recovery from End-of-Life NdFeB Permanent Magnet Scrap: A Critical Review. Journal of Sustainable Metallurgy, 2017. 3(1): p. 122-149, https://doi.org/10.1007/s40831-016-0090-4.
40. M. V. Reimer, H. Y. Schenk-Mathes, M. F. Hoffmann and T. Elwert., Recycling Decisions in 2020, 2030, and 2040—When Can Substantial NdFeB Extraction be Expected in the EU? Metals, 2018. 8(11): p. 867, https://doi.org/10.3390/met8110867.
41. Z. Wenga, N. Haque, G. M. Mudd, S. M. Jowitt, Assessing the energy requirements and global warming potential of the production of rare earth elements. Journal of cleaner production, 2016. 139: p. 1282-1297, https://doi.org/10.1016/j.jclepro.2016.08.132.
42. Goodenough, K.M., F. Wall, and D. Merriman, The rare earth elements: demand, global resources, and challenges for resourcing future generations. Natural Resources Research, 2018. 27(2): p. 201-216, https://doi.org/10.1007/s11053-017-9336-5.
43. Jin, H.Y., et al., Life Cycle Assessment of Neodymium-Iron-Boron Magnet-to-Magnet Recycling for Electric Vehicle Motors. Environmental Science & Technology, 2018. 52(6): p. 3796-3802, https://doi.org/10.1021/acs.est.7b05442.
44. Lalana, E.H., et al. Recycling of Rare Earth Magnets by Hydrogen Processing and Re-Sintering. in European Congress and Exhibition on Powder Metallurgy. European PM Conference Proceedings. 2016. The European Powder Metallurgy Association.
45. Rabatho, J.P., et al., Recovery of Nd and Dy from rare earth magnetic waste sludge by hydrometallurgical process. Journal of Material Cycles and Waste Management, 2013. 15(2): p. 171-178, https://doi.org/10.1007/s10163-012-0105-6.
46. M. Zakotnik, C. O. Tudor, L. T. Peiró, P. Afiuny, R. Skomski, G. P.Hatch, Analysis of energy usage in Nd–Fe–B magnet to magnet recycling. Environmental Technology & Innovation, 2016. 5: p. 117-126, https://doi.org/10.1016/j.eti.2016.01.002.
47. Diehl, O., et al., Towards an Alloy Recycling of Nd–Fe–B Permanent Magnets in a Circular Economy. Journal of Sustainable Metallurgy, 2018. 4(2): p. 163-175, https://doi.org/10.1007/s40831-018-0171-7.
48. A. Lixandru, I. Poenaru, K. Guth, R. K. Gauss, O. Gutfleisch, A systematic study of HDDR processing conditions for the recycling of end-of-life Nd-Fe-B magnets. Journal of Alloys and Compounds, 2017. 724: p. 51-61, https://doi.org/10.1016/j.jallcom.2017.06.319.
Reviewer 3 Report
Experimental methods, results and discussions are well organized.
Improvements in magnetic properties are clearly shown and the mechanism is discussed in derail.
Several inquiries and comments are attached.

Author Response
Reviewer #3:
Inquiries
- Regarding the introduction (line 48), is there any special reason to except the work below? K. Hirota, H. Nakamura, T. Minowa, M. Honshima: Coercivity Enhancement by Grain Boudnary Diffusion Process to Nd-Fe-B Sintered Magnets, IEEE Transactions on Magnetic 42 (10), pp. 2909 – 2911 (2006).
We thank the reviewer for posting this query. This work presented an alternative method back in 2006 related to the Grain Boundary Diffusion (GBD) process applicable to the Nd-Fe-B permanent magnets, therefore the reference was incorporated in the introduction part to acknowledge this thoroughly researched and industrially applicable technique. But it is also important to mention that the GBD process is limited to thickness of the magnets less than 5 mm and the diffusion is not uniform either from the surface towards the center. So the magnetic properties from these different zones will be different and the magnetic hardening (coercivity) is limited against thermal fluctuations or demagnetizing fields. Therefore, a more robust option is to blend the powders and achieve localized uniform diffusion profiles for the comprehensive core-shell microstructure development which we had exploited for the first time in case of the HDDR Nd-Fe-B system.
- What is the targeting Hci and Br? Is your results in optimized conditions enough for practical use?
From Ref [10], the recycled HDDR powder has HCi equivalent to the commercial grade MF-15P HDDR powder (1190 kA/m). The target is to approach the state of the art value of 1550 kA/m (µ0Ha ~ 2T) for the HDDR powder [13]. But in comparison to industrial requirement (from internal request in the project) for the Nd-Fe-B sintered magnets, the coercivity value of 1400 kA/m was considered adequate for testing up to 100 OC service temperature. Since our work is related to coercivity improvement of the recycled HDDR powder and isotropic magnets were made, the Br increase was not in focus. But it was important to understand how the magnetization changes occur with the dopant weight fraction and the thermal treatment conditions.
- Did you find any difference in phase formation/diffusion of Dy along with the annealing time by the SEM BSE observation? (Only results after 6-hours-annealing were shown in the article, though desirable properties were obtained after 1-hour-annealing.)
We acknowledge this excellent remark from the reviewer.
The prolonged thermal treatment conditions were chosen because of identifying the complex intermetallic phase and the diffusion of Dy from the edges of the HDDR powder particles. The microstructure for optimal conditions is similar to what is shown in Figure 5 but with the prolonged holding times, the core-shell structure begins to collapse for > 5 wt. % of dopant. The complex intermetallics were less abundant for the optimally heat treated samples at 1 hour, therefore it was important to understand why at longer periods diffusion of Dy stops (circumventing the core-shell microstructure) and the formation of RE-FX (RE= Dy, Nd & X = 3, 4), DyFe2 and Nd-Fe-F phases. The relationship of complex non-optimal microstructure with the magnetic properties were easy to explain when diffusion mechanism was studied for longer heat treatment conditions. This important information would have not been suitably identified since core-shells form throughout the sample even for 1-hour thermal treatment but the intermetallics grow out of the reaction zone over prolonged periods. These findings have been elaborated comprehensively in the discussion part.
Comments
- Letter on Figure 1(a) and some numbers on Figure 6 cannot be read when the manuscript is printed in black and white. Even in full-color printing, some numbers on the Figure 6 are not clear.
The Figure 1 has been modified to increase the legibility of text.
Likewise, the Figure 6 has been thoroughly modified for EDXS labels and text marks for better visibility and the clarity of the subject. This scheme is also readable in grey-scale mode.
Corrections
- Line 73: “The thermal ramp of 100 OC” à Do you mean the heating rate of 100 OC/min?
Thank you for notifying and in order to simplify the understanding, the sentence from line 74 – 77 has been rephrased according to the suggestion by reviewer:
“The heating rate of 100 OC was maintained till 700 OC and reduced to 50 OC/min for reaching the maximum temperature; 1 minute of holding time was given at 750 OC to reach nearly full densification. The sintering temperature was measured with a calibrated infrared pyrometer.”
- Line 91: Figure 1 à Figure 1 (a).
This has been adjusted in line 92 as suggested.
- Line 196: “this phase in reduced to NdF4” à “this phase is reduced to NdF4”
We apologize for the typing/grammatical mistake and line 197 now incorporates the suggested correction:
“this phase is reduced to NdF4”
- Line 246: “phase contains a now a lesser amount of Dy” à “phase contains now a lesser amount of Dy”
The sentence from line 260 – 263 has been modified to improve the legibility for the reader:
“Below this ternary transition temperature ~400 nm thick DyNdFe14B shells already have deposited on the surface of Nd2Fe14B grains and the solidifying liquid phase now contains a lesser amount of Dy but more concentration of Nd.”
